# Dietary Grape Seed Meal Bioactive Compounds Alleviate Epithelial Dysfunctions and Attenuates Inflammation in Colon of DSS-Treated Piglets

**DOI:** 10.3390/foods10030530

**Published:** 2021-03-04

**Authors:** Gina Cecilia Pistol, Cristina Valeria Bulgaru, Daniela Eliza Marin, Alexandra Gabriela Oancea, Ionelia Taranu

**Affiliations:** 1Laboratory of Animal Biology, INCDBNA-IBNA, National Institute of Research and Development for Biology and Animal Nutrition, 077015 Balotesti, Romania; cristina.bulgaru@ibna.ro (C.V.B.); daniela.marin@ibna.ro (D.E.M.); ionelia.taranu@ibna.ro (I.T.); 2Laboratory of Chemistry and Nutrition Physiology, INCDBNA-IBNA, National Institute of Research and Development for Biology and Animal Nutrition, 077015 Balotesti, Romania; alexandra.oancea@ibna.ro

**Keywords:** grape, inflammatory bowel disease, epithelium, inflammation, animal model

## Abstract

Inflammatory Bowel Diseases (IBD) are chronic inflammations associated with progressive degradation of intestinal epithelium and impairment of the local innate immune response. Restoring of epithelial integrity and of the mucosal barrier function, together with modulation of inflammatory and innate immune markers, represent targets for alternative strategies in IBD. The aim of our study was to evaluate the effects of a diet including 8% grape seed meal (GSM), rich in bioactive compounds (polyphenols, polyunsaturated fatty acids (PUFAs), fiber) on the markers of colonic epithelial integrity, mucosal barrier function, pro-inflammatory, and innate immunity in DSS-treated piglets used as animal models of intestinal inflammation. Our results have demonstrated the beneficial effects of bioactive compounds from dietary GSM, exerted at three complementary levels: (a) restoration of the epithelial integrity and mucosal barrier reinforcement by modulation of claudins, Occludin (*OCCL)* and Zonula-1 (*ZO-1)* tight junction genes and proteins, myosin IXB (*MYO9B)* and protein tyrosine phosphatase (*PTPN)* tight junction regulators and mucin-2 (*MUC2)* gene; (b) reduction of pro-inflammatory *MMP-2* (matrix metalloproteinase-2) and *MMP-9* (matrix metalloproteinase-9) genes and activities; and (c) suppression of the innate immune *TLR-2* (Toll-like receptor-2) and *TLR-4* (Toll-like receptor-4) genes and attenuation of the expression of *MyD88* (Myeloid Differentiation Primary Response 88)/*MD-2* (Myeloid differentiation factor-2) signaling molecules. These beneficial effects of GSM could further attenuate the transition of chronic colitis to carcinogenesis, by modulating the in-depth signaling mediators belonging to the Wnt pathway.

## 1. Introduction

Inflammatory bowel disease (IBD) is a chronic inflammation of the small intestine and/or the colon resulting from the alteration of intestinal homeostasis and of inappropriate mucosal immune activation [1]. The pathogenesis of IBD is still poorly understood, this disease being a multifactorial combination of genetic, environmental, microbial, and immunological factors. The chronic inflammatory responses in IBD lead to a progressive degradation of the intestinal epithelium (structure and function), which, in turn, favors more stimuli leakage. This excessive afflux of inflammatory stimuli (antigens, cytokines and others) exacerbates inflammation and compromises the integrity of the intestinal mucosal barrier, directing to progression and to an increased severity of IBD [2]. For example, pro-inflammatory cytokines (e.g., tumor necrosis factor (TNF)-α, interferon-γ) leads to an increased tight junction permeability and induces apoptosis of intestinal epithelial cells [3,4]. Furthermore, exposure of epithelial cells to TNF-α alters mucins production and disrupts the epithelial barriers [5].

The IBD pharmacological treatments are based on anti-inflammatory drugs (e.g., mesalazine), biologics acting as cytokine blockers (anti-TNF-α antibodies such as Adalinumab and Infliximab), steroids or immunosuppressants such as azathioprine and methotrexate [6]. Of these treatments, the anti-TNF blockers (e.g., infliximab, adalimumab, certolizumab pegol, and golimumab) have been the only approved biologic drugs for the treatment of IBD [7]. These therapies, however, can only alleviate symptoms and are associated with a number of potential risks, including infection (due to microbiota dysbiosis and proliferation of pathogenic bacteria), local reactions, and malignancy [8].

Recently, the attention has focused on the management of the damage and abnormal functioning of the intestinal epithelial barrier occurring in IBD and the role of tight junction components and modulators [9]. This management cannot be fully achieved by pharmaceutical approaches, and new alternative tools to restore the intestinal epithelium integrity and functions are needed [10].

The nutritional factors have aroused attention as alternative IBD treatment approaches, and their effectiveness is usually studied in animal models of experimental colitis [6]. Of these, alternative strategies based on bioactive compounds found in plants such as polyphenols (flavonoids, phenolic acids, stilbenes, and lignans), fatty acids (omega 3 and omega 6), and fibers demonstrated beneficial effects on intestinal epithelium damage and inflammation [11,12]. Polyphenols (resveratrol, curcumin, narigenin, naringin etc.) showed protective effects on epithelial barrier dysfunction during dextran sulphate sodium-(DSS) induced colitis in mice, through regulation of junction protein expression and improving also the innate immunity at intestinal level [13,14,15,16]. There are also studies focused on the role of polyunsaturated fatty acids (PUFAs) in reducing the signs of colitis. Most of these studies have demonstrated the role of ω-3 PUFAs in restoring the mucosal architecture and ulceration in both human IBD and in animal models (DSS-induced mouse colitis; 4,6-trinitrobenzene sulfonic acid- (TNBS-) induced rat colitis) [17,18]. Additionally, dietary fiber could affect the intestinal epithelial barrier functions by their inter-relation with intestinal microbiota [19]. For example, fermentable Guar Gum improved the expression levels of junction proteins in a murine model of DSS-colitis [20] and pre-treatment with psyllium fiber before DSS-inducing colitis in mice ameliorated epithelial dysfunction and the expression of extracellular matrix-associated genes [21].

The main grape by-products (pomace, seeds, and peels) represent a good source of active compounds, especially polyphenols that make them suitable to be considered as functional ingredients [22]. An increasing number of studies demonstrated the beneficial effects of grape compounds and grape by-products in various chronic diseases, including IBD [23,24]. In mice with DSS-induced colitis, grape seed proanthocyanidin extract alleviates colitis symptoms by modulating inflammatory cytokines and oxidative stress, maintaining the intestinal barrier, and by improving the microbial community [25]. Furthermore, prophylactic consumption of polyphenol-rich grape pomace extracts exerts protective effects against DSS-induced colitis, by decreasing the inflammatory status at the colon level [26]. The beneficial effects of polyphenols from grape extract in terms of improvement of the intestinal epithelial structure was observed also by Wang et al. [27] in a mouse model of colitis (IL-10 deficient mice).

Our previous studies demonstrated that the diet with 8% whole grape seed meal (GSM), containing high amounts of polyphenols, PUFAs and fiber, was efficient in counteracting DSS-induced colon inflammation by reducing pro-inflammatory cytokines and chemokines production and by attenuating mitogen-activated kinases (MAPKs) and nuclear factor kappa B (NF-*k*B) gene and protein expressions [28]. Furthermore, GSM diet positively modulated the microbiota in DSS-treated piglets, stimulating the growth of beneficial bacteria and the production of butyrate and isobutyrate short-chain fatty acids (SCFAs) [29]. We thus hypothesized that whole GSM, by its content in bioactive compounds, could improve also the epithelium integrity and functionality affected by the DSS challenge. Therefore, in the present study we investigated the effects of the diet with 8% GSM on the colonic epithelial integrity (junction proteins and regulators of tight junctions), on the mucosal barrier function, pro-inflammatory matrix metalloproteinases, and innate immunity markers in DSS-challenged piglets as models for IBD. Collectively, the role of whole GSM in the restoration of the epithelial integrity and functionality corroborated with previously reported anti-inflammatory and microbiota modulator effects that might provide more evidence for its utilization as promoter of intestinal health.

## 2. Materials and Methods

### 2.1. Experimental Diets

A control diet (compound feed—a corn-soybean meal basal diet) and a GSM meal diet (compound feed with inclusion of 8% grape seed meal) were used in this study. The diets respected the specific requirements for weaning piglets. At the beginning of the experiment, feed samples were collected and were analyzed for nutrient content. The Weende method was used to analyze the basic chemical composition of experimental diets (crude protein, crude fat, crude fibers, and ash), according to the ISO methods (ASRO-SR EN ISO, 2010). The grape seed meal (GSM) used in this study was provided as dried material by a local distributor (S.C. OLEOMET S.R.L., Bucharest, Romania). The diets composition in polyunsaturated fatty acids (PUFAs) was measured by gas chromatography as described by Taranu et al. [30]. Folin-Ciocalteu and HPLC-DAD-MS analyses were used to determine the total polyphenol content and different classes of polyphenols in experimental diets, as described by Taranu et al. [30]. The antioxidant capacity of experimental diets was determined as described by Garcia et al. [31].

### 2.2. Animals and DSS-Induced Colitis

There were four groups of weanling piglets (TOPIGS-40 crossbred piglets 21-days old, *n* = 5 per group/pen) studied for 30 days. The initial average body weight of piglets was 9.04 ± 0.13 kg. Animals were individually identified by ear tag. Piglets were acclimatized for one week before the experiment started and were housed in pens within the experimental base of the National Research-Development Institute for Animal Nutrition and Biology, Balotesti, Romania. The body weight was recorded on day 0 (the beginning of the experiment) and day 30 (the end of the experiment) for each animal. On every day of the experiment, assigned diet and water were provided ad libitum.

Piglets were randomly assigned to four equal experimental groups of *n* = 5 piglets/group (with comparable initial body weight). Animals were assigned to a control diet without GSM (1—Control group and 2—DSS group) and to an experimental diet including 8% GSM (3—GSM group and 4—DSS + GSM group)

The protocol of the induction of chronic colitis in piglets was described in Pistol et al. [28] and in Taghipour et al. [32]. Briefly, two cycles of DSS treatment (an oral dose of 1g/b.w/day of DSS (Dextran sulphate 40 sodium salt, MW = 36–50 kDa, Carl Roth GmbH, Karlsruhe, Germany) were performed in piglets from experimental groups 2 (DSS) and 4 (DSS + GSM), on days 1–5 and 21–25 of the experiment. All throughout the experimental period, piglets under DSS treatment were daily monitored for illness indicators: body-weight loss, diarrhea, and overall condition of the animal. All animals were handled in accordance with rules for handling and protection of animals used for experimental purposes (Romanian Law 206/2004 and decision 28/2011). At the end of the experiment, animals were slaughtered by exsanguination according to the EU Council Directive 2010/63/CE. The colon samples collected on ice from all animals were kept at −80 °C until the analyses. The study protocol was approved by the Ethical Committee of the National Research-Development Institute for Animal Nutrition and Biology (INCDBNA-IBNA), Balotesti, Romania.

### 2.3. Quantitative Polymerase Chain Reaction (qPCR) Analysis

Frozen colon samples (50 mg) were lysed in TRIzol Reagent (Sigma-Aldrich Chemie GmbH, Darmstadt, Germany) and homogenized using Ultra-Turrax homogenizer (IKA-Werke GmbH &Co. KG, Staufen, Germany). Chloroform-isopropanol method [28] was used for total RNA (tRNA) isolation. Complementary DNA (cDNA) was synthesized from tRNA samples using M-MLV Reverse Transcriptase kit (Life Technologies, Carlsbad, CA, USA) according to the manufacturer’s protocol.

For qPCR, a mixture of 25 ng cDNA template, 10 µL SYBR Green qPCR Master Mix (Life Technologies, Carlsbad, CA, USA), and nuclease-free water (to a final volume of 20 µL) was added to each well containing lyophilized primers, according to manufacturer‘s protocol. The sequences of primer pairs used in this study are presented in the Appendix A. The cycling protocols and selection of reference genes were performed as described by Pistol et al. [28]. For qPCR data normalization, the Excel-based NormFinder software was used for the selection of the two reference genes (ACTB coding for β-actin, and PPIH for peptidyl-prolyl cis-trans isomerase H). Results were expressed as relative fold change (Fc) compared with control group.

### 2.4. Immunoblot Analysis of Junction Protein Expression

The expression levels of Claudin-4, Occludin and Zonula-1 (ZO-1) proteins were analyzed by Western blotting. The colon samples were homogenized and lysed using a modified RIPA buffer as described in Pistol et al. [28]. Then, 30 μg of total colon proteins were separated on a 10% SDS-PAGE and transferred onto nitrocellulose membrane. The membranes were probed with rabbit anti-Claudin-4, Occludin and ZO-1 antibodies (ThermoFisher Scientific, Rockford, IL, USA; diluted 1/1000) and β -actin antibody (Cell Signaling Technology, Danvers, MA, USA; diluted 1/500). After three steps of washing, the blots were incubated for 1 h at room temperature with a horseradish peroxidase-conjugated goat anti-rabbit antibody (1:2000, Cell Signaling Technology, Danvers, MA, USA). The antibody detection was performed as previously described [28].

### 2.5. Gelatin Zymography

The determination of the gelatinolytic activities of MMP-2 and MMP-9 in colon was realized by zymography using the protocol described in Pistol et al. [33]. Briefly, frozen colon samples of 50 mg were lysed in TBS lysis buffer, on ice. The total protein content in lysate samples was determined using a commercial kit (Pierce BCA Protein Assay Kit, ThermoFisher Scientific, Rockford, IL, USA). The lysates were aliquoted and frozen at −80 °C until processing. 30 µg of samples were separated onto an SDS-PAGE electrophoresis (10% polyacrylamide gel containing 0.1% gelatin). The proteins were further renatured by incubation of gels for 30 min in 2.5% Triton-X100 solution and the gelatinolytic activity was developed by incubation of the gels in the enzymatic substrate at 37 °C for 18 h. The gels were stained with Coomassie Brilliant Blue and scanned. The enzymatic activities of MMPs were quantified using GelQuant software (DNR Bio-Imaging Systems LTD, Jerusalem, Israel). All densitometry results were expressed as arbitrary units (AU).

### 2.6. Immunofluorescence Analysis of Junction Proteins

Caco-2 cells, grown onto cell imaging slides (Eppendorf GmbH, Hamburg, Germany), at a concentration of 1 × 10^5^ cells/mL, were treated for 4 h with 5 µg/mL of LPS (lipopolysaccharide) for the induction of inflammation, as described in Pistol et al. [34]. After LPS treatment, cells were cultured in the presence of GSM extract (50 µg/mL gallic acid equivalents GAE) for 24 h at 37 °C in a 5% CO2 humidified atmosphere. The cells were fixed with PBS (phosphate buffered saline) containing 3.7% paraformaldehyde (30 min at room temperature) and blocked with 10% BSA (bovine serum albumin) in PBS (30 min, room temperature). Samples were incubated (overnight, at 4 °C) with anti-Claudin-4 and anti-Occludin antibodies (ThermoFisher Scientific, Rockford, IL, USA), diluted 1:100. Slides were washed with washing buffer and incubated with secondary fluorescent-conjugated antibodies (Anti-Rabbit IgG (H+L), CF488A and Anti-Rabbit IgG (H+L), CF594 antibody, Sigma-Aldrich Chemie GmbH, Schnelldorf, Germany), diluted 1/100. For nuclear staining, Hoechst 33,342 (Cell Signaling Technology, Inc, Danvers, MA, USA) 1 µg/mL was included in the last 15 min of the secondary antibody incubation step. Images were captured using a Nikon Ts2RFL inverted microscope. Images were captured using a CCD camera and processed using ZEN 3.2 (Blue edition) (Carl Zeiss Microscopy GmbH, Oberkochen, Germany) or ImageJ (Fiji version) [35] software. There were three independent experiments performed.

### 2.7. Statistical Analysis

The results were presented as mean ± standard error of the mean (SEM). One-way ANOVA and Student’s t-test tests, followed by Fisher’s procedure of the least square difference, were used for all pairwise comparison between experimental groups (StatView software 6.0, SAS Institute, Cary, NC, USA), and *p* < 0.05 was considered statistically significant. The homogeneity of the results was assessed by Pettitt’s test and Buishand’s test for homogeneity (XLSTAT2020 software).

qPCR results of 27 genes (belonging to junction markers group, MMPs/TIMPs group, TLRs group and Wnt signaling group) in all experimental groups were analyzed with principal component analysis (PCA). The principal component analysis (PCA) statistical multivariate method (XLSTAT2020 software) was used also for cluster analysis of experimental groups.

## 3. Results

### 3.1. The Composition of GSM Diet

The composition of experimental diets (Control and diet containing GSM) has already been described by Pistol et al. [28]. Briefly, the GSM diet had a high content in total polyphenols compared to the Control diet (897.15 mg GAE/100 g vs. 382.93 mg GAE/100 g), among these flavonols (catechin, epigallocatechin), isoflavonoids (daidzein derivatives), and anthocyanins (procyanidin, petunidin, malvidin) are in important concentrations. This makes their antioxidant activity also higher (966.3 μM TRE/g vs. 206.89 μM TRE/g). The GSM diet had also an increased content of ω-6 fatty acids (50.56 g/100 g FAME) and of fiber (5.89%) compared to the Control diet (ω-6 fatty acids: 45.38 g/100 g FAME and 3.56%, respectively).

### 3.2. Effect of GSM Diet on DSS-Induced Colitis

As shown in Figure 1, the average body weight decreased in DSS-challenged piglets measured at the end of the experiment, when compared to the Control group (18.8 ± 1.8 kg in DSS group, vs. 22.9 ± 1.5 kg in Control group, Figure 1). By contrast, piglets with DSS-induced colitis receiving the GSM diet recovered the control level. No difference between piglets from Control and GSM groups in terms of final body weight was found (Figure 1).

### 3.3. Effect of GSM Diet on Colonic Endothelial Integrity under the In Vivo DSS Action

#### 3.3.1. Effect on the Colon Tight Junction Genes Expression

The levels of the expression of genes coding for several tight junction proteins (surface claudins: *CLDN1* (Claudin-1), *CLDN4* (Claudin-4), *CLDN23* (Claudin-23); crypt claudins: *CLDN2* (Claudin-2), *CLDN5* (Claudin-5), *CLDN14* (Claudin-14) and *CLDN20* (Claudin-20); occludin *OCCL* and zonula-1 *ZO-1*) with a key role in the trans-cellular transport of molecules were analyzed by qPCR, in order to evaluate the effects of DSS treatment and the restorative capacity of GSM diet on colon epithelial junctions/epithelium integrity. Our qPCR results presented in Table 1 clearly showed that DSS treatment dramatically downregulated 89% (8/9) of the analyzed junction mRNAs in colon, the most affected genes being those coding for *CLDN23* surface claudin (−74% reduction, *p* = 0.006 vs. Control, Table 1) and *CLDN20* crypt marker (−78% reduction, *p* = 0.042 vs. Control, Table 1). The gene coding for *CLDN2* is upregulated (3-fold increase) by DSS challenge (Table 1). The GSM diet restored all junction genes affected by DSS treatment to the Control level (Table 1). On the other hand, GSM treatment alone did not produce any significant effects compared with the control.

#### 3.3.2. Effect on the Colon Tight Junction Proteins Expression

Further, the qPCR results were validated by immunoblot analysis presented in Figure 2. Indeed, DSS treatment decreased significantly the expression of junction proteins compared to the Control group (CLDN4: −34% reduction, *p* = 0.001 vs. Control; OCCL: −38% reduction, *p* = 0.004 vs. Control; ZO-1: −31% reduction, *p* = 0.001 vs. Control, Figure 2). Importantly, the addition of GSM in the diet effectively counteracted the effect induced by DSS by increasing all analyzed junction proteins in the colon, compared to the DSS group (CLDN4: +34% increase, *p* = 0.044 vs. DSS group; OCCL: +64% increase, *p* = 0.006 vs. DSS; ZO-1: +31% increase, *p* = 0.031 vs. DSS, Figure 2). The treatment with GSM alone did not produce significant effects when compared with the control.

To further address the effectiveness of GSM in alleviating the inflammation-induced changes at intestine level, we used an in vitro established Caco-2 cell culture system stimulated with LPS as a cellular model of intestinal inflammation [34]. Briefly, we examined the effects of a GSM extract on the claudin 4 CLDN4 and occludin OCCL immunostaining intensity in LPS-treated Caco-2 cells. Immunofluorescence microscopy images presented in Figure 3 showed that LPS treatment strongly reduced the intensity of the staining for both CLDN4 and OCCL in Caco-2 cells (CLDN4: −55%, *p* = 0.010 vs. Control cells; OCCL: −42%, *p* < 0.001 vs. Control, Figure 3). The addition of GSM extract to LPS-treated cells increased the expression level of CLDN4 over the LPS level (Figure 3), but this expression was decreased when compared to the Control (Figure 3). In cells treated with GSM extract alone, the CLDN4 expression was not affected compared to the Control (Figure 3). The treatment of LPS-stimulated Caco-2 cells with GSM extract restored the OCCL expression toward the Control level, whereas the addition of GSM extract alone did not affect OCCL expression compared to Control (Figure 3).

#### 3.3.3. Effect of GSM Diet on Junction Regulation Genes in Colon of DSS-Treated Piglets

Analysis of genes coding for tight junction regulation showed that DSS treatment induced a strong downregulation of 80% (4/5) of these mRNAs compared to the control group, the most affected being *MAGI2* (Membrane Associated Guanylate Kinase, WW And PDZ Domain Containing 2) and *GNAI2* (Guanine nucleotide-binding protein G(i) subunit alpha-2) genes (*MAGI2*: −90% decrease, *p* < 0.001 vs. Control, *GNAI2*: −60% reduction, *p* = 0.035 vs. Control, Table 1). The inclusion of GSM in the diet of piglets exposed to DSS was not able to restore the gene level of *MAGI2* and *GNAI2* toward the control, but increased the *MIO9B* mRNA, coding for myosin IXB protein, over the DSS and even the Control level (5.4-fold increase, *p* < 0.001 vs. DSS group; 2-fold increase, *p* = 0.019 vs. Control group, Table 1). In the colon of piglets challenged with DSS, a significant downregulation of the *PTNP2* gene involved in tight junction regulation was found compared to the Control (−60% decrease, *p* = 0.020 vs. Control, Table 1). GSM diet was efficient in counteracting the effect of DSS by increasing the gene expression of *PTPN2* toward that of Control level (0.9 ± 0.2Fc, *p* = 0.021 vs. DSS, Table 1). The Cadherin 1 *CDH1* mRNA (coding for epithelial cadherin) was unmodified in all experimental groups (Table 1).

### 3.4. Effects of GSM on DSS-Induced In Vivo Inflammation in Colon

The aim of this study was to evaluate the effect of GSM dietary inclusion against DSS-induced inflammation. The impact of GSM diet on several important genes and proteins involved in inflammation was analyzed. 

#### 3.4.1. Effect on the Gene’s Expression of Extracellular Matrix Degradation

The expression of two important metalloproteinases genes, *MMP-2* (matrix metalloproteinase-2) and *MMP-9* (matrix metalloproteinase-9), and their tissue inhibitors, *TIMP-1* (tissue inhibitor of matrix metalloproteinases 1) and *TIMP-2* (tissue inhibitor of matrix metalloproteinases 2), with a key role in the regulation of extracellular matrix degradation and known to be involved in inflammatory conditions of the colon were analyzed in order to assess the potential of GSM diet against DSS-induced inflammation. The qPCR analysis showed that DSS increased the mRNA levels of both *MMP-2* (1.67-fold increase, *p* = 0.008 vs. Control, Figure 4A) and *MMP-9* (2.49-fold increase, *p* = 0.023 vs. Control, Figure 4A) and by contrast, a reduction of both *TIMP-1* and *TIMP-2* gene expressions in the colon of the DSS-treated group (*TIMP-1*: −48% reduction, *p* = 0.018 vs. Control; *TIMP-2*: −64% reduction, *p* < 0.001 vs. Control, Figure 4A). The GSM diet increased the gene expression for *TIMPs* and *MMPs* over the DSS level, restoring them to the Control level (Figure 4A).

#### 3.4.2. Effect on Metalloproteinases Activity

The results of gelatin zymography analysis showed that both MMP-2 and MMP-9 activities were increased in the colon of DSS-treated piglets compared to the Control group (+108% increase in MMP-2 activity, *p* = 0.001 vs. Control; +139% increase in MMP-9 activity, *p* = 0.003 vs. Control, Figure 4B). Similar with qPCR data, the effect of GSM diet was the restoring of both MMP-2 and -9 activity to the Control level (Figure 4B).

#### 3.4.3. Effect of GSM Diet on DSS Induced Damage to Barrier Function

The colonic mucosa provides a vital defensive barrier and the barrier integrity defects increase the translocation of bacterial antigens, stimulating inflammation of the intestinal mucosa. These damages were associated in IBD with defective mucin production or processing [2]. Of barrier constituents, mucin2 (MUC2) is the major colonic mucin affected in IBD. The extracellular matrix 1 protein (ECM1) is required for the intestinal barrier formation in intestinal epithelium [2]. To study the effects of GSM diet on the damage induced by the DSS challenge, we analyzed qPCR *MUC2* and *ECM1* gene expression. Our results showed that DSS treatment reduced the expression of genes coding for *MUC2* and *ECM1* in colon (*MUC2*: −78% decrease, *p* = 0.002 vs. Control; ECM1: −83% reduction, *p* = 0.010 vs. Control, Figure 5). GSM diet increased the *MUC2* mRNA toward the Control level in the colon of DSS-treated piglets (0.84 ± 0.24 Fc, *p* = 0.023 vs. DSS group, Figure 5). GSM alone had no effect on *MUC2* gene expression and induced an insignificant downregulation of *ECM1* mRNA (0.6-fold decrease, *p* > 0.050) compared to the Control (Figure 5).

### 3.5. The Effect of GSM Diet on Wnt Signaling Genes Expression in Colon of DSS-Treated Piglets

The Wingless/Int1 (Wnt) signaling pathway is a key regulator of epithelial proliferation, and its activation is a feature of chronic colitis [36]. Therefore, we evaluated in colon samples the effects of the DSS and GSM diet on the gene expressions of *FZD1* (Frizzled-1), *FZD3* (Frizzled-3), *WNT2B* (Wnt Family Member 2B) and *WNT5B* (Wnt Family Member 5B) members of this pathway. DSS treatment induced an increase of 3/4 of the Wnt signaling genes. The most affected gene was *FZD3*, with a 3-fold increase compared to the Control (*p* = 0.012, Figure 6). By contrast, *FZD1* mRNA level was downregulated in the colon of DSS-treated piglets (0.45-fold decrease, *p* = 0.034 vs. Control, Figure 6). GSM diet counteracted the DSS effects, by decreasing the gene expression of *FZD3*, *WNT2B* and *WNT5B* under the Control level (Figure 6). Furthermore, GSM diet increased the *FZD1* mRNA level over that from Control (Figure 6).

### 3.6. Effect of GSM Diet on Colonic Innate Immunity under In Vivo DSS Action

#### 3.6.1. Effect on Toll-Like Receptors (TLRs) Gene Expression

Toll-like receptors (TLRs) represent key mediators of innate immunity at intestinal level, their dysfunction being associated with the pathogenesis of colitis [37,38]. In our experiment, a percentage of 50% (5/10) of colonic TLRs genes were upregulated by DSS treatment (Table 2). Among these, *TLR-2, TLR-4*, and *TLR-6* genes were the most affected genes in the DSS group (*TLR-2*: 3.1-fold increase, *p* = 0.004 vs. Control; *TLR-4*: 2.1-fold increase, *p* = 0.037 vs. Control; *TLR-6*: 2.4-fold increase, *p* = 0.012 vs. Control, Table 2). All TLRs genes upregulated by DSS treatment were restored toward the Control levels by GSM diet (Table 2). GSM diet alone did not affect the TLRs gene expression compared to Control (Table 2).

#### 3.6.2. Effect on TLRs Signaling Adapter Genes

To assess the impact of the GSM diet on in-depth cellular response to DSS, we further analyzed the gene expression of several adapter molecules genes (*MyD88, MD-2,* Nucleotide oligomerization domain *NOD,* Interleukin-1 receptor-associated kinase 1 *IRAK1,* TNF Receptor Associated Factor 6 *TRAF6* and Toll-interacting protein *Tollip*). The qPCR analysis of the in-depth TLR signaling mediators showed that DSS challenge induced an upregulation of all analyzed TLR adapter genes (6/6) in the colon (Table 3). Of these, *MyD88* and *MD-2* genes had the most increased expression in DSS group, compared to Control (*MyD88*: 6.8-fold increase, *p* = 0.001 vs. Control; *MD-2*: 6.9-fold increase, *p* < 0.001 vs. Control, Table 3). Grape seed meal diet significantly reduced all analyzed TLR signaling genes under the level from DSS group (*NOD2*, *p* = 0.007 vs. DSS, *MyD88*, *p* = 0.002 vs. DSS and *MD-2*, *p* < 0.001 vs. DSS, Table 3) or toward Control level (*IRAK1*, *p* = 0.019 vs. DSS, *TRAF6*, *p* = 0.010 vs. DSS and *Tollip*, *p* = 0.010 vs. DSS, Table 3). GSM diet alone also increased the level of *NOD2* mRNA over the Control level (2.3-fold increase, *p* = 0.007, Table 3), while *Myd88*, *MD-2*, *IRAK1*, *TRAF6* and *Tollip* genes were unmodified by GSM alone compared to Control (Table 3).

### 3.7. Principal Component Analysis (PCA)

To evaluate the changes of gene expressions and their correlation with experimental groups, the qPCR data set on 27 markers were subjected to PCA analysis. The genes belong to the main classes of markers analyzed in this study: junction markers (*CLDN1*, *CLDN2*, *CLDN4*, *CLDN5*, *CLDN14*, *CLDN20*, *CLDN23*, *OCCL* and *ZO-1*), MMPs/TIMPs (*MMP-2, MMP-9, TIMP-1* and *TIMP-2*), Wnt signaling (*FZD1, FZD3, WNT2B, WNT5B*), and TLRs (*TLR1, TLR2, TLR3, TLR4, TLR5, TLR6, TLR7, TLR8, TLR9, TLR10*). The PCA clearly distinguished the DSS experimental group (separated in the first principal component) from Control, GSM and DSS+GSM groups (Figure 7A). Interestingly, the DSS+GSM group cluster is closely related to the Control cluster (Figure 7A). The genes which contribute to these differences are shown in Figure 7B. Analysis of DSS group revealed a high expression of both *MMP-2* and *MMP-9* genes, *CLDN*2 and Wnt associated mRNAs. These genes strongly differentiated the DSS group from the Control group. On the other hand, the distribution of genes coding for junction proteins, TIMPs and TLRs, showed a higher expression of these genes in DSSS+GSM group (Figure 7B) and were overlapped with genes overexpressed in the Control group.

## 4. Discussion

In our recent paper [28], we demonstrated the capacity of the GSM diet to positively modulate the expression of inflammatory markers (cytokines, chemokines) in the colon of young piglets used as models for a DSS-induced colitis. An increased level of pro-inflammatory cytokines leads to the degradation of colon epithelium, mediated by activated MMPs, a family of proteolytic enzymes including MMP-2 and -9 gelatinases [39,40,41]. In the inflamed intestinal tissues collected from IBD patients, MMPs appeared at a high level, suggesting that these enzymes have important roles in initiation and perpetuation of intestinal ulceration and inflammation [42]. Furthermore, colonic MMP-2 and -9 levels have been closely correlated with the disease severity and chronic damage of the colon in DSS-treated animals [26,43]. In the present study, *MMP-2* and *-9* gene expressions as well as their activities were stimulated by DSS challenge (Figure 4A,B), while the expression of their inhibitors *TIMP-1* and *TIMP-2* were reduced in the colon of DSS-treated piglets (Figure 4A). The inclusion of 8% of GSM in the diet restored the *MMPs* levels (Figure 4A,B) and increased the gene expression for MMPs inhibitors *TIMP-1* and *TIMP-2* in the colon of DSS-treated piglets (Figure 4A). Similarly, dietary supplementation with polyphenol-rich extracts from different sources of grapes pomace [26] and with polyphenols from grape peel powder [44] reduced the upregulation of MMP-9 expression in DSS-challenged rats and in TNBS-treated rats, respectively. The inhibition of DSS-induced expression of MMP-2 and MMP-9 inflammatory mediators by the whole GSM supported our previous findings showing that the diet with 8% grape seed meal with its bioactive compounds could counteract intestinal inflammation induced by DSS [28].

Epidemiologic, genetic, and clinical results and data from in vivo and in vitro models of IBD supported the hypothesis that intestinal homeostasis altered in IBD is also affected by an impaired innate immune response [37]. Thus, in colitis, individual TLRs, usually expressed on macrophages and dendritic cells with a key role in innate immunity and with differential expression at intestinal level [37,38]. These authors reported that *TLR-2*, *-4*, *-8*, and *-9* genes are overexpressed in active UC, whereas TLR5 is downregulated in quiescent UC (compared to control patients) [38]. Of all TRLs, *TLR-2* and *TLR-4* were associated with acute colitis and with progression of intestinal tissue destruction for further ulceration, leading to promotion of intestinal inflammation [45,46]. In the study herein, the DSS challenge led to the upregulation of 50% (5/10) of TLRs genes; of these, *TLR-2* and *TLR-4* were the most affected by DSS treatment (Table 2). Inclusion of 8% GSM in the piglet diet had a restorative effect on the TLR genes in GSM+DSS group compared to the DSS group. There are both in vitro and in vivo studies showing the effects of polyphenols and ω3-PUFAs on *TLR-2* and *TLR-4* in different models of inflammation. For example, Yang et al. [47] showed that EGCG ((−)-epigallocatechin-3-gallate), a green tea polyphenol, downregulates *TLR-4* expression and blocks NF-kB activity in LPS-stimulated intestinal cell line IEC-6 and suggests that it may be effective as an anti-inflammatory factor in IBD. Furthermore, by using LPS-stimulated RAW264.7 macrophages, HEK293 kidney cells and LPS-challenged female Balb/c mice, Shibata et al. [48] demonstrated the inhibition of TLR-2 and TLR-4 proteins by iberin and quercetin (polyphenols from cabbage and onion). All these effects demonstrated that TLR-2 and TLR-4 could be the anti-inflammatory target of polyphenols. Apple polyphenol phloretin had the same effect in the stimulated RAW264.7 macrophage cell line, which also demonstrated the role of phenolic compounds in the modulation of inflammatory processes [49]. The diet supplementation with grape peel powder extract rich in polyphenols reduced TLR-4 expression in the colon of TNBS- treated rats, attenuating the inflammatory and oxidative response [44]. Additionally, ω3-PUFAs have an inhibitory effect on the expression of TLR4 in LPS-challenged piglets [50].

It was shown that the TLR signaling pathways are involved in the progression of IBD, mainly via MyD88/MD2-dependent pathways [51]. Indeed, our results demonstrated that the DSS challenge increased the levels of all analyzed TLR signaling genes (*MyD88*, *MD-2*, *NOD*, *IRAK1*, *TRAF6* and *Tollip*), especially for *MyD88* and *MD-2* (Table 3). By contrast, the diet with GSM significantly reduced all analyzed TLR adapter genes under the level of those from the DSS group (Table 3). It was demonstrated that extracted polyphenols or phenolic synthetic compounds could modulate the TLR signaling mediators, inhibiting in cascade the TLR-induced inflammation. *TLR-4*, *MyD88* or *TRAF6* gene expression or protein were inhibited by either quercetin (polyphenol found in grapes and red wine), curcumin (a yellow polyphenols from Curcuma longa plants) or resveratrol (polyphenol found in grapes and cocoa) in LPS-stimulated RIMVECs, RAW264.7, and HEK293 cells (quercetin or iberin), inflamed colon of TNBS-treated rats (curcumin), liver and lung of LPS-treated rats (resveratrol) and LPS-challenged female Balb/c mice (iberin and quercetin) [48,52,53,54].

The increased expression of pro-inflammatory cytokines and of innate immune mediators (TLRs and MyD88 adaptor molecule) influence further the integrity of the epithelial barrier in colitis [55,56,57]. The initiation and progression of intestinal inflammation in IBD is associated with the damage of epithelial barrier integrity [58]. Literature data have demonstrated that DSS administration induces epithelial barrier dysfunction by internalization and degradation of junction proteins, leading to the destabilization of intercellular junctions in animal models [59,60,61,62]. As expected, also in our study, DSS challenge induced a significant downregulation of the colon tight junction marker genes, *CLDN4*, *CLDN5*, *CLDN14*, *CLDN20*, *CLDN23*, *OCCL* and *ZO-1* (Table 1). On the other hand, both in human IBD [63] and in DSS-challenged animals [64] increased levels of claudin 2 were found and correlated with the severity of colitis. Similarly, in our experiment, *CLDN2* gene was strongly upregulated in the colon of DSS-treated piglets. The GSM diet showed a real capacity to restore all protein junction genes affected by DSS treatment (Table 1). Furthermore, a similar effect of GSM diet was found at protein level: CLDN4, OCCL and ZO-1 proteins overexpressed in DSS group were restored to the control level in DSS-treated piglets by GSM diet (Figure 2). Other studies pointed out the protective effects of polyphenols, fiber and PUFAs on gut integrity and function, both in IBD patients and animal models of colitis [44,65]. For example, red wine phenolic extract diminished the expression of key tight junction proteins OCCL, ZO-1 and CLDN5 in cytokine-stimulated HT-29 cell line [66]. In mice model of colitis, resveratrol [62], diet with red raspberries powder (rich in insoluble fiber) [59], and dietary CLA (conjugated linolenic acid) administration upregulated tight junction proteins *E-cadherin 1, ZO-1, CLDN3* and *OCCL* gene expressions [60]. In rats with TNBS-induced colitis, diet supplementation with grape peel powder rich in polyphenols and fiber [44] or with fish oil rich in ω-3 PUFAs [65] strongly upregulated the ZO-1, CLDN1, CLDN5 and CLDN8 protein expressions in colon. By contrast, dietary ω-3 and ω -6 PUFAs did not affected tight junction proteins CLDN1 and OCCL in the colon of TNBS-treated rats [67]. The junction protein expression restoring effect of GSM could be assigned to its complex composition in bioactive compounds, whole GSM being a valuable candidate for epithelial integrity and maintenance.

In IBD, *MAGI2*, *GNAI2*, *MIO9B*, *PTPN2*, and *CDH1* genes have been identified as tight junction assembly and barrier function regulators and were significantly associated with colitis [68]. In our study, DSS treatment reduced significantly the expression of 80% (4/5) of these genes. Dietary GSM was able to increase 2/5 of the tight junction genes assembly (40%), *MYO9B* and *PTPN* genes over or toward the Control level (Table 1). Based on our knowledge, this is the first in vivo study investigating some tight junction assembly genes in an animal model of colitis, even if these genes were directly associated with IBD [68]. Moreover, only one meta-analysis showed that coffee polyphenols, caffeic acid and chlorogenic acid increased the *MYO9B*, *PTPN11* and *CDH1* gene expressions in un-stimulated colon adenocarcinoma HT29 cells [69].

The degradation of the extracellular matrix is associated with the defects of mucosal barrier, contributing to the propagation and persistence of inflammatory processes. Moehle et al. [70] demonstrated that, in ulcerative colitis, changes in the expression of *MUC2* gene, the most abundant mucin from the colon, was correlated with the decrease in mucus layer thickness. Indeed, in our experiment the DSS reduced strongly (−78%) the mRNA level for colonic *MUC2* gene, and this effect was counteracted by the dietary GSM (Figure 5). There are few studies demonstrating the effects of grape polyphenols on mucins expressions in animal models of IBD. For example, grape seed extract supplementation was associated with enhanced mRNA expression of *MUC2* in IL10-deficient mice model of colitis [71]. Furthermore, the diet supplemented with DMW (phenolic extract of dealcoholized muscadine wine) increased levels of secreted fecal mucin 2 in DSS-treated mice [72].

The activation of homeostatic Wingless/Int1 (Wnt) pathway has been observed in chronic colitis both in IBD patients and animal models [36,73]. Additionally, the dysregulation of the Wnt pathway is a common event in transition from colitis to colon carcinogenesis [74]. A microarray study showed that genes coding for Wnt ligands and for Frizzled receptors exhibited either significantly increased (*WNT2B*, *WNT3A*, *WNT5B*, *WNT6*, *WNT7A*, *WNT9A*, *WNT11*, *FZD3* and *FZD4*) or decreased (*FZD1* and *FZD5*) gene expression in ulcerative colitis compared to non-IBD patients [75]. Our results showed that DSS treatment induced an upregulation of 75% (3/4) of the analyzed genes belonging to Wnt pathway (*FZD3*, *WNT2B* and *WNT5B*), while the gene expression for *FZD1* was significantly downregulated (Figure 6). Diet with 8% GSM restored all these genes toward the control level in GSM+DSS group (Figure 6). In the same way, studies reporting the positive effects of dietary resveratrol on Wnt pathway members in DSS-induced colitis, restoring Wnt signaling in rat colon [76] and downregulating the protein expression of WNT5A both in spleen and colon of mice model were reported [77]. Moreover, it was shown that the anti-carcinogenic properties of resveratrol are due to the inhibition of the Wnt signaling in HCT 116 colon cancer cells [78].

## 5. Conclusions

In conclusion, our results clearly show that GSM, by its composition in bioactive compounds such as polyphenols, PUFAs, fiber and others, is effective in restoring epithelial integrity and the mucosal barrier reinforcement by restoring tight junction genes and proteins, tight junction regulators and *MUC2* genes. Corroborating with our previous findings [27], the reduction of pro-inflammatory MMPs gene expression and activity by GSM diet indicates a significant anti-inflammatory effect of this by-product against intestinal inflammation and epithelial disorders. This intestinal anti-inflammatory action is also related to the suppression of innate immunity markers TLR-2 and TLR-4 and by attenuation of the increased expression of MyD88/MD-2 signaling pathway members. Furthermore, these beneficial effects of GSM could further attenuate the transition of chronic colitis to carcinogenesis, by modulating the in-depth signaling mediators belonging to Wnt pathway.

## Figures and Tables

**Figure 1 foods-10-00530-f001:**
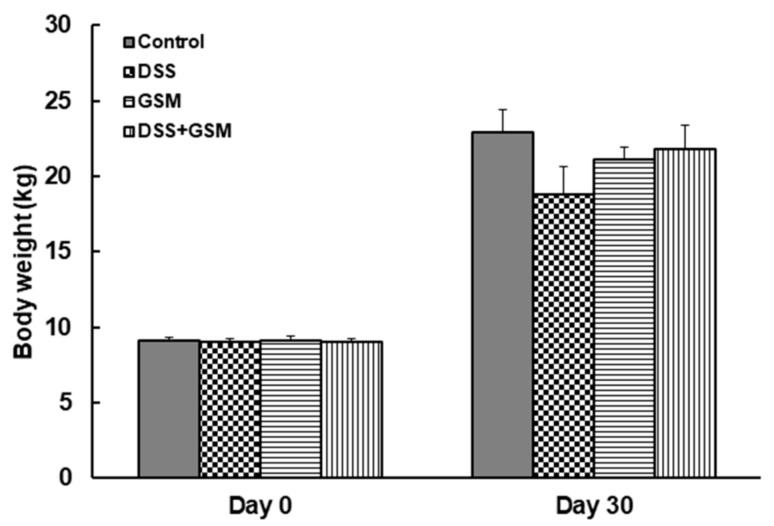
The effects of grape seed meal (GSM) diet on body weight evolution; unchallenged and dextran sulphate sodium (DSS)-treated pigs were assigned for 30 days to a control diet (Control and DSS groups) or 8% GSM diet (GSM and DSS + GSM groups). The body weight was recorded at the beginning and at the end of the experiment. The results, expressed are presented as mean ± SEM.

**Figure 2 foods-10-00530-f002:**
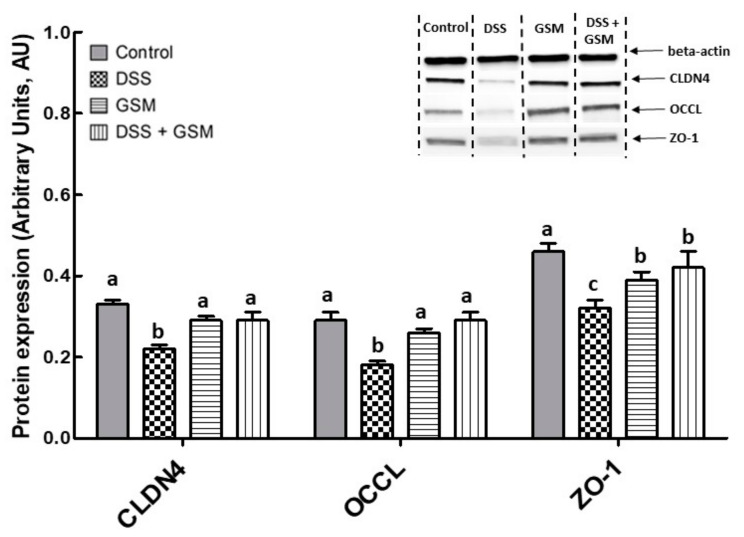
The effect of dietary treatments on tight junction protein expression in colon lysates. Unchallenged and DSS-treated pigs were assigned for 30 days to a control diet (Control and DSS groups) or 8% GSM diet (GSM and DSS + GSM groups). At the end of the experiment, colon samples from all animals (*n* = 5/group) were collected and Claudin 4, occludin, and Zonula-1 expression levels were determined by using Western blot analysis. The tight junction protein expressions were normalized as ratio to β-actin band intensities. Results are expressed as arbitrary units (A.U.) and presented as means ± standard errors of the mean (SEM). Anova-one way followed by Fisher tests were used to analyze the effect of experimental treatments on protein expression. ^a,b,c^ Groups with unlike superscript letters were significantly different (*p* < 0.05).

**Figure 3 foods-10-00530-f003:**
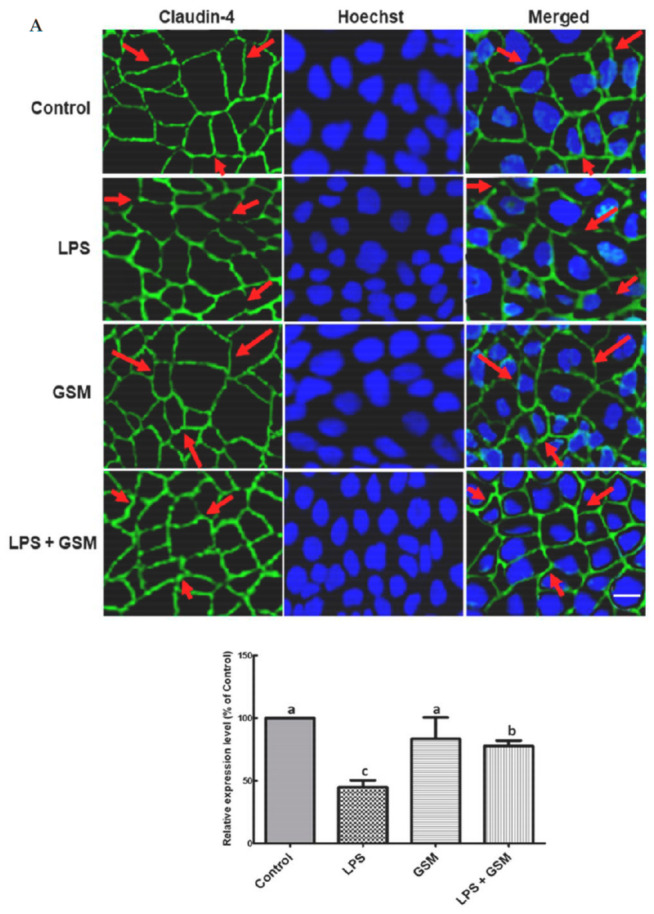
Identification of tight junction proteins in LPS-treated Caco-2 by immunofluorescence staining of CLDN4 (**A**) and OCCL (**B**). Confluent Caco-2 monolayers were treated for four hours with LPS or media (control), followed by 24-h incubation with GSM extract, as described in the Materials and Methods section. At the end of experiment, cells were stained with CLDN4 (green), OCCL (red) for junction proteins visualization, and with Hoechst (blue) for nuclei detection. Merged images show both tight junction proteins and nuclei staining. The arrows showed the intact junction protein expression (in Control, GSM and LPS + GSM samples) and deregulated expression of junction proteins (in LPS-treated cells), respectively. Scale bar = 20 µm. The charts from the right indicate the analysis of CLDN4 and OCCL relative fluorescence intensity using Image J (Fiji), expressed as percentages of Control samples. Anova-one way followed by Fisher tests were used to analyze the effect of experimental treatments on protein expression. ^a,b,c^ Groups with unlike superscript letters were significantly different (*p* < 0.05).

**Figure 4 foods-10-00530-f004:**
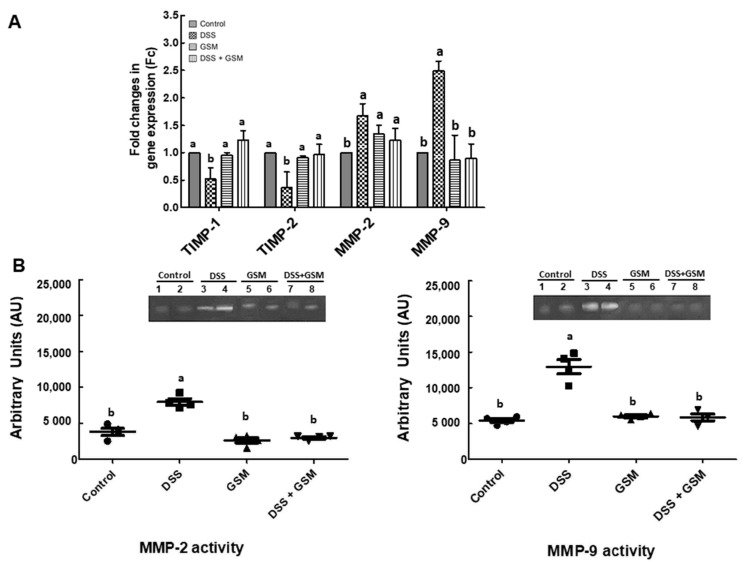
The effects of dietary treatments on the gene expressions of matrix metalloproteinases (MMP) and tissue inhibitors of MMPs (TIMP) (**A**) and on the activity of matrix metalloproteinases (MMPs) (**B**) in the colon of DSS-treated piglets. (**A**) Colon samples were analyzed for the TIMPs/MMPs gene expressions. The expression of target gene was normalized to the geometric mean of two reference genes; the results, expressed as fold changes (Fc) are presented as mean ± SEM. Values are the means from two independent replicates of colon samples (n = 5/group), with their standard errors represented by vertical bars. ^a,b^ Groups with unlike superscript letters were significantly different (*p* < 0.05; one-way ANOVA followed by Fisher test). (**B**) Colon lysates were resolved on SDS–PAGE gelatin zymography as described in Materials and Methods. Results were expressed as arbitrary units (A.U.) and are represented as means ± SEM. ^a,b^ Groups with unlike superscript letters were significantly different (*p* < 0.05; one-way ANOVA followed by Fisher test).

**Figure 5 foods-10-00530-f005:**
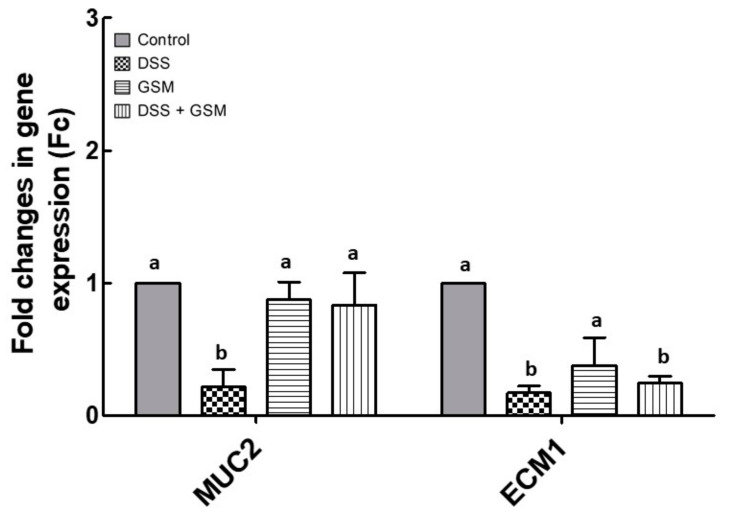
The effects of dietary treatments on genes are associated with barrier function damage in colon. *MUC-2* and *ECM1* gene expressions were evaluated in the colon of DSS-treated piglets by qPCR. The expression of target gene was normalized to the geometric mean of two reference genes; the results, expressed as fold changes (Fc) are presented as mean ± SEM. Values are the means from two independent replicates of colon samples (*n* = 5/group), with their standard errors represented by vertical bars. ^a,b^ Groups with unlike superscript letters were significantly different (*p* < 0.05; one-way ANOVA followed by Fisher test).

**Figure 6 foods-10-00530-f006:**
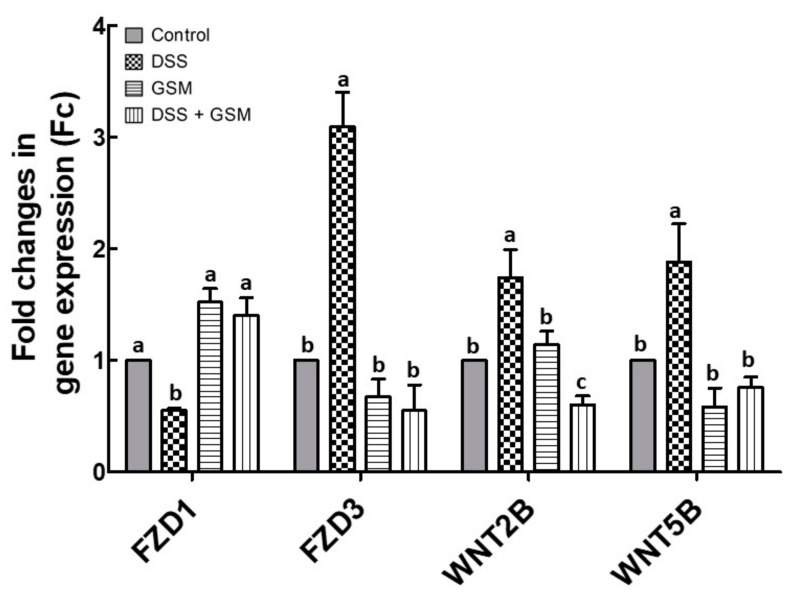
The effects of dietary treatments on genes coding for Wnt signaling members in the colon. *FZD1*, *FZD3*, *WNT2B* and *WNT5B* gene expressions were evaluated in the colon of DSS-treated piglets by qPCR. The expression of target gene was normalized to the geometric mean of two reference genes; the results, expressed as fold changes (Fc) are presented as mean ± SEM. Values are the means from two independent replicates of colon samples (*n* = 5/group), with their standard errors represented by vertical bars. ^a,b,c^ Groups with unlike superscript letters were significantly different (*p* < 0.05; one-way ANOVA followed by Fisher test).

**Figure 7 foods-10-00530-f007:**
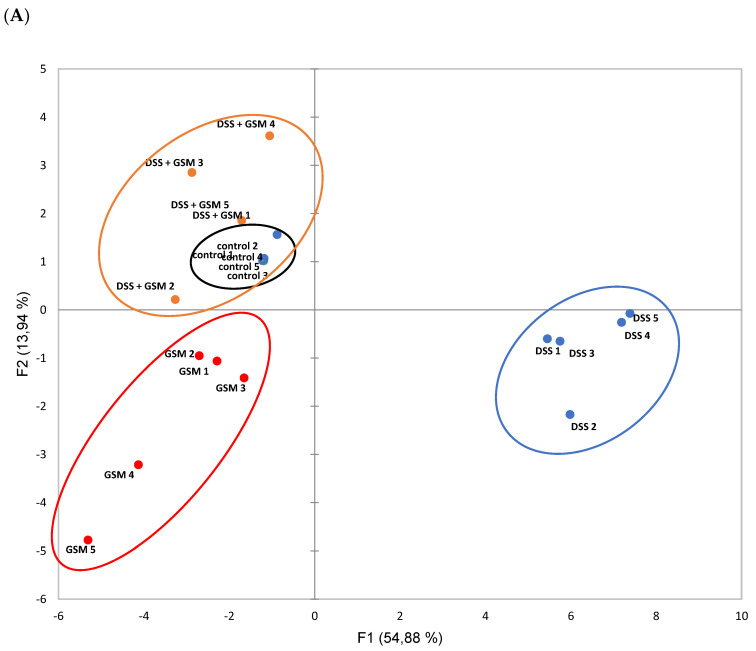
Principal component analysis (PCA) plots experimental groups with 27 genes based on qPCR data set. (**A**). The score plot differentiates the four experimental groups by two-dimensional expression clustering. Each point represents one sample; the color indicated sample groups, as follows: black color—Control group, blue color—DSS group, red color—GSM group and orange color—DSS + GSM group. (**B**). The loading plot with differentially expressed genes in relation to the largest portion of the variance among experimental groups.

**Table 1 foods-10-00530-t001:** The effects of GSM diet on the level of junction marker mRNAs in the colon of DSS-treated piglets.

Functional Classes of Genes	Gene	Experimental Group *							
		Control	DSS		GSM		DSS + GSM		
		Mean ± SEM **	Mean ± SEM **	Regulation vs. Control ^‡^	Mean ± SEM **	Regulation vs. Control ^‡^	Mean ± SEM **	Regulation vs. Control ^‡^	Regulation vs. DSS ^‡^
Junction proteins	*CLDN1*	1.0 ± 0.0 ^a^	0.4 ± 0.2 ^b^	down	0.9 ± 0.2 ^a^	−	0.9± 0.1 ^a^	−	up
	*CLDN2*	1.0 ± 0.0 ^b^	3.2 ± 0.3 ^a^	up	1.3 ± 0.3 ^b^	−	0.9 ± 0.1 ^b^	−	down
	*CLDN4*	1.0 ± 0.0 ^a^	0.4 ± 0.1 ^b^	down	0.7 ± 0.2 ^a^	−	0.8 ± 0.1 ^a^	−	up
	*CLDN23*	1.0 ± 0.0 ^a^	0.2 ± 0.0 ^b^	down	0.9 ± 0.2 ^a^	−	1.0 ± 0.3 ^a^	−	up
	*CLDN5*	1.0 ± 0.0 ^a^	0.3 ± 0.0 ^b^	down	0.9 ± 0.1 ^a^	−	0.9 ± 0.2 ^a^	−	up
	*CLDN14*	1.0 ± 0.0 ^a^	0.3 ± 0.0 ^b^	down	1.0 ± 0.1 ^a^	−	1.3 ± 0.4 ^a^	−	up
	*CLDN20*	1.0 ± 0.0 ^a^	0.2 ± 0.0 ^b^	down	1.5 ± 0.2 ^a^	−	1.0 ± 0.3 ^a^	−	up
	*OCCL*	1.0 ± 0.0 ^a^	0.3 ± 0.0 ^b^	down	0.7 ± 0.2 ^a^	−	1.0 ± 0.3 ^a^	−	up
	*ZO-1*	1.0 ± 0.0 ^a^	0.3 ± 0.0 ^b^	down	1.5 ± 0.2 ^a^	−	1.0 ± 0.2 ^a^	−	up
Markers of tight junction regulation	*MAGI2*	1.0 ± 0.0 ^a^	0.1 ± 0.0 ^b^	down	0.3± 0.2 ^b^	down	0.1 ± 0.0 ^b^	down	−
	*GNAI2*	1.0 ± 0.0 ^a^	0.4 ± 0.2 ^b^	down	1.1 ± 0.3 ^a^	−	0.6 ± 0.1 ^b^	down	−
	*MIO9B*	1.0 ± 0.0 ^b^	0.4 ± 0.1 ^c^	down	1.4 ± 0.3 ^b^	−	2.0 ± 0.3 ^a^	up	up
	*PTPN2*	1.0 ± 0.0 ^a^	0.4 ± 0.1 ^b^	down	0.7 ± 0.3 ^a^	−	0.9 ± 0.2 ^a^	−	up
	*CDH1*	1.0 ± 0.0 ^a^	0.7 ± 0.3 ^a^	−	1.0 ± 0.2 ^a^	−	0.8 ± 0.1 ^a^	−	−

* Unchallenged and DSS-treated pigs were assigned for 30 days to a control diet (Control and DSS groups) or 8% GSM diet (GSM and DSS + GSM groups). At the end of the experiment, colon samples from all animals (*n* = 5) were collected and analyzed for gene expression by qPCR. ** the expression of target gene was normalized to the geometric mean of two reference genes; the results, expressed as fold changes (Fc) are presented as mean ± SEM. ^‡^ Anova-one way followed by Fisher tests were used to analyze the effect of experimental treatments on mRNA expression, and *p* < 0.05 was considered statistically significant; up: up-regulation, down: down-regulation, -: no effect. ^a,b,c^ Groups with unlike superscript letters were significantly different (*p* < 0.05).

**Table 2 foods-10-00530-t002:** The effects of GSM diet on Toll-like receptors (TLRs) gene expression in the colon of DSS-treated piglets.

Gene	Experimental Group *							
	Control	DSS		GSM		DSS + GSM		
	Mean ± SEM **	Mean ± SEM **	Regulation vs. Control ^‡^	Mean ± SEM **	Regulation vs. Control ^‡^	Mean ± SEM **	Regulation vs. Control ^‡^	Regulation vs. DSS ^‡^
*TLR-1*	1.0 ± 0.0 ^a^	1.5 ± 0.2 ^a^	−	1.4 ± 0.1 ^a^	−	1.3 ± 0.3 ^a^	−	−
*TLR-2*	1.0 ± 0.0 ^b^	3.1 ± 0.4 ^a^	up	1.4 ± 0.1 ^b^		1.2 ± 0.2 ^b^	−	down
*TLR-3*	1.0 ± 0.0 ^b^	2.0 ± 0.3 ^a^	up	1.2 ± 0.1 ^b^	−	0.9 ± 0.2 ^b^	−	down
*TLR-4*	1.0 ± 0.0 ^c^	2.1 ± 0.4 ^a^	up	1.8 ± 0.3 ^b^	up	0.9 ± 0.3 ^c^	−	down
*TLR-5*	1.0 ± 0.0 ^a^	1.8 ± 0.4 ^a^	−	0.9 ± 0.2 ^a^	−	0.9 ± 0.2 ^a^	−	−
*TLR-6*	1.0 ± 0.0 ^b^	2.4 ± 0.3 ^a^	up	0.7 ± 0.1 ^b^	−	1.1 ± 0.3 ^b^	−	down
*TLR-7*	1.0 ± 0.0 ^a^	1.1 ± 0.1 ^a^	−	1.1 ± 0.1 ^a^	−	1.0 ± 0.3 ^a^	−	−
*TLR-8*	1.0 ± 0.0 ^b^	3.4 ± 0.1 ^a^	up	1.0 ± 0.1 ^b^	−	1.5 ± 0.2 ^b^	−	down
*TLR-9*	1.0 ± 0.0 ^a^	1.1 ± 0.3 ^a^	−	1.8 ± 0.2 ^a^	−	1.4 ± 0.1 ^a^	−	−
*TLR-10*	1.0 ± 0.0 ^a^	1.1 ± 0.2 ^a^	−	1.3 ± 0.2 ^a^	−	1.3 ± 0.2 ^a^	−	−

* Unchallenged and DSS-treated pigs were assigned for 30 days to a control diet (Control and DSS groups) or 8% GSM diet (GSM and DSS + GSM groups). At the end of the experiment, colon samples from all animals (*n* = 5) were collected and analyzed for gene expression by qPCR. ** the expression of the target gene was normalized to the geometric mean of two reference genes; the results, expressed as fold changes (Fc) are presented as mean ± SEM. ^‡^ Anova-one way followed by Fisher tests were used to analyze the effect of experimental treatments on mRNA expression, and *p* < 0.05 was considered statistically significant; up: up-regulation, down: down-regulation, -: no effect. ^a,b,c^ Groups with unlike superscript letters were significantly different (*p* < 0.05).

**Table 3 foods-10-00530-t003:** The effects of GSM diet on the expression of TLRs signaling-associated genes in the colon of DSS-treated piglets.

Gene	Experimental Group *							
	Control	DSS		GSM		DSS + GSM		
	Mean ± SEM **	Mean ± SEM **	Regulation vs. Control ^‡^	Mean ± SEM **	Regulation vs. Control ^‡^	Mean ± SEM **	Regulation vs. Control ^‡^	Regulation vs. DSS ^‡^
*NOD2*	1.0 ± 0.0 ^c^	5.7 ± 0.6 ^a^	up	2.3 ± 0.4 ^b^	up	2.5 ± 0.5 ^b^	Up	down
*MyD88*	1.0 ± 0.0 ^c^	6.8 ± 0.6 ^a^	up	1.2 ± 0.4 ^c^	−	2.1 ± 0.4 ^b^	Up	down
*MD-2*	1.0 ± 0.0 ^c^	6.9 ± 0.4 ^a^	up	0.9 ± 0.1 ^c^	−	4.1± 0.4 ^b^	Up	down
*IRAK1*	1.0 ± 0.0 ^b^	2.7 ± 0.3 ^a^	up	1.3 ± 0.3 ^b^	−	1.0 ± 0.1 ^b^	−	down
*TRAF-6*	1.0 ± 0.0 ^b^	4.8 ± 0.3 ^a^	up	1.0 ± 0.1 ^b^	−	0.8 ± 0.3 ^b^	−	down
*Tollip*	1.0 ± 0.0 ^b^	1.5 ± 0.3 ^a^	up	1.2 ± 0.2 ^b^	−	0.9 ± 0.1 ^b^	−	down

* Unchallenged and DSS-treated pigs were assigned for 30 days to a control diet (Control and DSS groups) or 8% GSM diet (GSM and DSS + GSM groups). At the end of the experiment, colon samples from all animals (*n* = 5) were collected and analyzed for gene expression by qPCR. ** the expression of target gene was normalized to the geometric mean of two reference genes; the results, expressed as fold changes (Fc) are presented as mean ± SEM. ^‡^ Anova-one way followed by Fisher tests were used to analyze the effect of experimental treatments on mRNA expression, and *p* < 0.05 was considered statistically significant; up: up-regulation, down: down-regulation, -: no effect. ^a,b,c^ Groups with unlike superscript letters were significantly different (*p* < 0.05).

## Data Availability

The data presented in this study are available on request from the corresponding author. The data are not publicly available due to policy of the National Research-Development Institute for Animal Nutrition and Biology, Balotesti, Romania.

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
