# Peer review of "Dietary Grape Seed Meal Bioactive Compounds Alleviate Epithelial Dysfunctions and Attenuates Inflammation in Colon of DSS-Treated Piglets"

_foods, 2021, doi:10.3390/foods10030530_

Round 1

Reviewer 1 Report

The manuscript deals with grape seed meal as dietary compound during experimental inflammatory diseases. The authors treated piglets with grape seed meal and the animals underwent DSS colitis. After that they isolated colon tissue and analysed the epthelial effector molecules ex vivo as well as mRNA expression and fluorescence immunostaining of tight junction molecules and transcription factors. Finally they found beneficial effects of grape seed meal in inflammatory reaction afflecting the barrier function of the mucosal layer in colon.

First, the authors used the DSS colitis model in the classic way, that means one cycle with 30 days. After that the analysis of the colon tissue was done. I would be careful to propagate that GSM is powerful in the „chronic colitis“, because only one cycle was performed. For using the term „chronic“ more cycles would be necessary. If you used only one cycle then it is a „acute“ inflammation and not a so called „chronic“ inflammation.

Concerning the inflammatory reaction mediated by DSS, it would be better to give the reader more information like a weight curve or HE staining of the colon tissue, to illustrate more the severity of the inflamed status. So please add a „inflammatory figure“.

To analyse the effect of the tight junction proteins then they used qPCR analysis with claudins, occludins and Zonula-1 mRNA molecules in colon tissue from the animals. For me it is unclear why they ahve focussed them just to only a small number of molecules like CLDN-4, OCCL and ZO-1, as there are some more interesting tight junction molecules. From literature it is known that during inflammatory bowel disease especially claudin-1, -5 are downregulated and claudin-2 mRNA molecules are up regulated. It would be nice to see a link between more molecules in the DSS model, too.

Finally, i am not really happy with the figure2. Why have the authors used Caco-2 cells for staining of claudin-4 and occludine and not the colon tissue of the animals? The immunohistochemical staining are not really persuadingly. A magnification bar or magnification hint is missing. The visible area is too far away from the cell-cell contacts and a higher magnnification is needed to see the differences. As a reference for a nice illustration with claudins the manuscript from Wiechmann et al., PLoS One, 2014 Nov 20;9(11) can be taken. For a better visualization some arrows would be helpful to focus the junctions as well.

Overall, the manuscript is solide, but it will need some more work to improve the quality of the presented figures. For that i propose a major Revision.

Reviewer 2 Report

This study aimed to analyse the dietary grape seed meal bioactive compounds to alleviate Inflammatory Bowel Diseases. As the Authors wrote in the manuscript, new alternative tools to restore the intestinal epithelium integrity and functions are needed because pharmaceutical approaches are no ideal.  Important is analyse the nutritional factors as alternative IBD treatment approaches.

Therefore, in my opinion, this study has high originality and significance. Moreover, the manuscript was prepared very well, but need revision.

  • The Introduction Section explains the design of the study. The Authors very well justify the research topic.
  • The study was carried out without methodological errors.
  • The Descriptions of the results were correct.
  • The presented figures and table were prepared precisely and also legible.
  • The Discussion Section includes the accurate reference of the results obtained to the studies of other studies. However, I have a small comment for the Discussion Section; I think that the chapter's first paragraph is unnecessary (lines 425-428) and please delete it.
  • The Conclusions were well formulated.

Reviewer 3 Report

The manuscript is very interesting.

My comments are following:

  1. The aim is not clearly written in the abstract.
  2. More exact findings from the obtained results should be added to the abstract too.
  3. The following reference should be added since it is emphasizing grape byproducts: Antonić, B., Jančíková, S., Dordević, D., & Tremlová, B. (2020). Grape Pomace Valorization: A Systematic Review and Meta-Analysis. Foods9(11), 1627.
  4. How the homogeneity of the results was evaluated, it should be added to the Method part.
  5. Why principal component analysis was not used?
  6. Lines 424-428: this paragraph is out of context!!!

Round 2

Reviewer 1 Report

The authors has answered all the critical points. For me the manuscript is now in a better status.